# Investigation into EHR data coverage in the All of Us Research Program via linkage to health insurance claims

Yuyang Yang[1]*, Kelsey Rodriguez[2], Javier Ezcurra[3], Romain Bogaerts[3], Andres Corrada-Emmanuel[3], Lew Berman[4], Melissa Basford[2], Abel Kho[1]

1 Feinberg School of Medicine, Northwestern University, Chicago, Illinois, United States of America, 2 Vanderbilt Institute for Clinical and Translational Research, Vanderbilt University Medical Center, Nashville, Tennessee, United States of America, 3 Swoop, New York, New York, United States of America, 4 All of Us Research Program, National Institutes of Health, Bethesda, Maryland, United States of America

* yuyang.yang@northwestern.edu

## Abstract

In 2020, the *All of Us* (AoU) Research Program Data Completeness Task Force identified several challenges in the state of program health data, including the high likelihood that Electronic Health Record (EHR) data from recruitment institutions is missing a significant portion of care received by participants. To improve data availability, the AoU Data Completeness Task Force recommended efforts to work with industry partners to better understand the degree of missingness within AoU EHR data, with an initial focus on claims data. In this study we describe efforts from AoU's collaboration with Swoop, a commercial analytics company holding health insurance claims data for over 80% of Americans, to assess the degree to which health insurance claims data can fill out the complete picture of care received by AoU participants. Using record linkage to link between individual participant AoU EHR data and their respective insurance claims, we quantitatively assess the amount of missing health data in both Swoop claims data and AoU EHR data over a decade long sample, identifying trends in data missingness based on participant characteristics. Our analysis demonstrates that AoU would greatly benefit from ingestion of claims data, gaining an estimated 16 million (90 per person) unique diagnosis codes, 17.8 million (99 per person) unique procedure codes, and 9.4 million (53 per person) unique drug codes through linkage to claims data.

## Introduction

### The *All of Us* Research Program

The *All of Us* (AoU) Research Program is an NIH-funded longitudinal cohort research study which aims to recruit a diverse sample of one million individuals to support the study of human health and advance precision medicine [1]. Participants in AoU agree

**Data availability statement:** All of the data used for analysis of this work can be found in our workspace on the All of Us user workbench. This link is now provided (https://workbench.researchallo-fus.org/workspaces/aou-rw-f6117030/duplicateofevaluatevisitspermonth/about).

**Funding:** "This study was supported by the National Institutes of Health in the form of a grant awarded to M.B. (1OT2OD035404-01) and Vanderbilt University Medical Center in the form of a salary for M.B. The specific roles of this author are articulated in the 'author contributions' section. The funders had no role in study design, data collection and analysis, decision to publish, or preparation of the manuscript".

**Competing interests:** I have read the journal's policy and the authors of this manuscript have the following competing interests: Abel Kho is an advisor of the company Datavant.

to share their health information, including electronic health record (EHR) data from care institutions, self-reported survey data about health behaviors, physical measurements, data from wearable devices like Fitbits, and genetic sequencing information with the program, providing a rich resource for health researchers. Currently the program has enrolled over 850,000 individuals across 50 US States and territories as of November 2024 [2].

## Care fragmentation within AoU EHR data

EHR data exists as a longitudinal record of patient exposures and outcomes, serving as a powerful source of real-world data to drive biomedical discovery that has become widely accepted in the study of diverse topics like healthcare utilization, medication post-marketing safety surveillance, and increasingly in clinical research [3]. Despite the promise of EHR data, issues involving data missingness and quality are known pitfalls of EHR data that need to be addressed to maximize its potential [4]. Recently internal investigations within AoU have identified significant potential for missingness in the shared EHR data of program participants [5–7]. The primary method for *All of Us* to acquire participant EHR data is from the healthcare provider organization (HPO) that a participant enrolls into the program through. For individuals that enroll into the AoU outside of an HPO (participants referred to as Direct Volunteers (DV)), EHR data can instead be ingested through connections to clinician portals, using Fast Health Interoperability Resource (FHIR) protocols, from healthcare institutions identified by the participant themselves [1,7]. In either case, after receiving participant consent and a HIPAA authorization to share clinical data [7,8], EHR data is converted to a standard format under the Observational Medical Outcomes Partnership (OMOP) common data model [9] before it is sent to the AoU Data and Research Center (DRC) [10]. Because participant EHR data typically comes from the enrollment HPO, the degree to which participant healthcare records outside of the enrollment HPO are included in AoU is limited. Due to this, a major potential source of EHR data missingness in AoU is care fragmentation [11], in which care received by patients at a healthcare institution is not necessarily communicated to the EHRs of other healthcare institutions the patient receives care at, leading EHR data from enrollment HPOs to provide an incomplete record of health activities [12].

To investigate the risk of care fragmentation in AoU EHR data, members of our group previously used a privacy-preserving record linkage method [13] to identify the proportion of shared patients between seven AoU-contributing HPO sites in three mid-western states (Wisconsin, Illinois, Indiana) between the years 2011–2018 [14]. Instances when the same individual showed up in more than one of these HPOs was identified at a rate of 6.1% to 32.7% between sites, suggesting significant potential for care fragmentation in patient records of AoU contributing HPO sites.

## Use of insurance claims to supplement EHR data

Health insurance claims are records of billable services submitted by healthcare systems to public or private insurers for the purposes of reimbursement. They contain standardized billing codes representing patient diagnoses, types of procedures

performed, and medications received. Due to their structured nature and availability, health insurance claims are frequently used in healthcare studies [15]. However, because claims data are designed for obtaining reimbursement, they may not contain non-billable services or conditions, under-document diagnoses not relevant to obtaining compensation, and incentivize documentation to make patients appear sicker ("upcoding") [16]. In addition, patients with unstable or no insurance may not be traceable using this datatype [16,17].In comparison, EHR data is primarily used for supporting and documenting patientcare. EHR data contains broader healthcare information than claims, including patient data from clinical notes, lab results, and medical history that are important for clinical care but not required for billing. While EHR data is also considered a valuable resource in health research, it has known data quality issues [18], as well as previously mentioned problems with care fragmentation.

It is largely supported that integration of claims and EHR data can be helpful in health research by providing a more complete picture of a patient's health. For example, use of combined data from both claims and EHR has been shown to outperform EHR or claims data alone in predictive modeling tasks across multiple scenarios [19–21]. While it may be helpful to link claims data to existing EHR systems, due to difficulties in interoperability, a lack of standardized pipelines to link EHR and claims data, and legal barriers surrounding HIPAA as well as state confidentiality laws, this practice is not typical [22,23].

In order to directly assess the extent of EHR data missingness within AoU participant data, we previously used record linkage to compare EHR data of 400,000 + AoU participants with claims data [24,25]. Unfortunately, only 41% of AoU participants matched to claims data provided by our data partner, Swoop, a precision health omnichannel solutions company with aggregated claims data for over 80% of Americans (obtained through HIPAA-compliant business associate agreements with primary sources including pharmacy benefit managers, clearinghouses, and payer organizations) [24]. Despite this limitation, our study identified significant missingness in the service dates, diagnosis events, procedure codes, and prescriptions in AoU EHR data compared to claims. In this manuscript, we describe improvements to our previous comparison of AoU EHR data with Swoop claims. We improved the participant match rate from 41% to 95% and conducted additional analysis to examine if participant level characteristics are associated with the extent of EHR data missingness when compared to claims.

## Materials and methods

Permission to conduct this study was obtained from the AoU Research Program Research Compliance Branch Institutional Review Board and Northwestern University Institutional Review Board. A cohort of 246,128 total AoU participants were included in the final analysis. AoURP participants provided written informed consent to share their EHR data with the AoURP for research purposes through a HIPAA authorization form, including linkage of their data to secondary data sources such as insurance claims [7,8]. EHR and claims data for each participant was included between the years 2011 and 2021. Data analysis took place between September 24th, 2021, and April 30th, 2025. All participant data was fully anonymized prior to access by authors, who did not have access to information that could identify individual participants during or after data collection.

### Data linkage process

Identifiers for each participant were aggregated by an analyst within the AoU DRC and include participant date of birth, full name, and gender. These identifiers were combined into a single string and used to create tokens at the individual person level to encrypt participant identity via software provided by the company Datavant. Using a one-way hash, two tokens for each participant were created from a string made up of the following patient identifiers [26].

1. $Token1 = LastName + 1st\ Initial\ of\ First\ Name + Sex\ at\ Birth + Date\ of\ Birth$

2. $Token2 = LastName\ (Soundex) + FirstName\ (Soundex) + Sex\ at\ Birth + Date\ of\ Birth$

After tokenization, the hashed identifiers from the AoU DRC, alongside counts of healthcare activity to be compared, are sent to Swoop, which generates their own tokens on the same set of identifiers. Swoop identifies persons that exist in both datasets, where a match exists when both Token1 and Token2 are the same between AoU and Swoop data. For each matching participant, Swoop calculates counts of healthcare activity from claims data on their side and sends the file back to AoU. Finally, the data is analyzed by the DRC team at Northwestern Medicine (NM) on the AoU Researcher Workbench.

## Description of shared data

We identified four healthcare activity types used for comparison in this study. 1) Service Dates, 2) Diagnosis (Dx) Codes (ICD9 and ICD10), 3) Procedure (Px) Codes (CPT and HCPCS), 4) National Drug Codes (NDCs).

**Patient event months.** We defined units of healthcare activity at the monthly level in the form of patient-event months (PEM). For a service day, each day in the calendar month in which healthcare activity (a Dx, Px, or NDC code) is reported adds one to the activity count within that PEM. For PEM counts in Dx, Px, NDC categories, the total number of reported activities in that month will be registered. For example, for the month of January 2018, if a participant has six diagnosis codes on three separate days within that month, a record of three will be recorded as the PEM count for service days regardless of the number of distinct diagnosis codes that occurred on those days for that patient, while a PEM count of 6 will be recorded for diagnoses in that month.

## Participant characteristics

We queried the AoU Researcher Workbench to identify additional information for each participant. These include participant self-identifying race, ethnicity, gender, and current age (by decade) and common chronic conditions. Chronic conditions were chosen based on the US Department of Health and Human Services Office of the Assistant Secretary of Health (OASH) list of prevalent chronic conditions [27] and include (Hypertension, Congestive Heart Failure, Coronary Artery Disease, Cardiac Arrhythmias, Hyperlipidemia, Stroke, Arthritis, Asthma, Autism Spectrum Disorder, Cancer, Chronic Kidney Disease, COPD, Dementia, Depression, Diabetes, Inflammatory disease of liver, HIV, Osteoporosis, Schizophrenia and Substance abuse disorder).

Additional patient data that is not available on the AoU Researcher Workbench (RWB) was obtained using an internal program data resource called the AoU Program Data Repository (PDR). This resource is not available on the RWB for general researcher use; however, it was used specifically on this project to inform programmatic activities and in terms of assessing EHR data quality (missingness) and determining the utility of pursuing claims data as an asset for use on the RWB. These included HPO Type (type of site through which the participant enrolls), income level, education, insurance type, and participant home address. We used the participant home Census tract to link in indices of socioeconomic disadvantage and degree of urbanization in the participant living area by linking to the Neighborhood Atlas Area Deprivation Index (ADI) [28,29] and Rural-Urban Commuting Area (RUCA) [30]. We used the national rank score of each census tract in the ADI, which provides a composite measure of deprivation between 1–100 and divided the score into quartiles (q1 [1–25], q2 [25–50], q3 [50–75], q4 [75–100]). RUCA codes were grouped into four categories depending on secondary RUCA code using "Scheme 1" provided by the Washington State Department of Health, which was chosen to account for participant accessibility to urban-based healthcare services [31]. Census tracts were categorized as urban core (1), suburban (2 and 3), large rural (4, 5, 6), and small town/rural (7, 8, 9, 10) depending on what number their respective RUCA codes started with.

## Statistical analysis

Descriptive statistics were calculated to identify differences in counts of healthcare activities seen between AoU and Swoop sets of data. Additionally, to see what patient characteristics were associated with differential activity counts between AoU and Swoop data, we performed an adjusted linear regression analysis using ordinary least squares

regression in the statsmodels package (Version 0.14.2) in Python. The covariates used were participant characteristics including demographic information (i.e., age, race, gender), socioeconomic characteristics (i.e., income, education, insurance type, deprivation index, geography type), and chronic conditions (i.e., heart failure, cancer, dementia). The reference values for this analysis were white (for race), non-hispanic (for ethnicity), male (for gender), 20–29-year-old (for age), annual income of 200k or greater (for income), advanced degree (for education), private (for insurance), urban core (for geography type), first quartile (being least deprived, for deprivation index) and not having the condition for all chronic diagnoses of interest. A cutoff of $P < 0.05$ is used to assess significance. All covariates used in the model had a variance inflation factor of less than 10. Because moderate heteroscedasticity of residuals was observed in the sample, we used robust standard errors measurements when calculating the regression.

Statistical analysis was performed using Python, version 3.10.12 within the *All of Us* Research Workbench (*All of Us* Registered Tier Dataset v7). A copy of our Jupyter Notebook and a link to our workspace is available upon request.

## Results

### Linkage evaluation

Health records for 472,877 AoU participants were initially considered for matching (Fig 1). Of these, 2,167 (.5%) participant records encountered a Datavant tokenization error (inability to generate tokens from patient identifiers), while 16,594 (3.5%) participants became ineligible for matching due to having identifiers that generate the same token pair as one or more other individuals. Of the 454,116 remaining potentially matchable individuals, 430,803 matched to a patient in the Swoop dataset, leading to a match rate of 95%. 429,995 individuals were available for analysis on the workbench, after finding that 808 (0.2%) of matched participants could not be identified in the AoU Researcher Workbench. We found that a substantial portion of AoU participants (178,331; 41.5% of the remaining participants) have yet to have EHR data ingested into the program (due to participants not providing HIPAA authorization to share EHR data with the program, a lag in the receipt of participant EHR data from partner HPOs, or that the participants joined the program digitally with no available EHR data) and were therefore removed from the primary analysis. A small portion of participants (n = 9) were then removed due to lack of data in demographic variables. Finally, participants without address information or those that failed geocoding were filtered out, leaving a final analysis cohort of 246,128 AoU participants (52% of the initial set).

When comparing the final analysis cohort (n = 246,128) to the set of participants that were missing EHR data but had valid demographic data on the AoU Researcher Workbench, no major discrepancies in the breakdown of demographic variables before and after exclusions were noted (Table 1).

### Analysis of recorded healthcare activity in EHR vs claims

The counts of healthcare activity found in AoU EHR data and Swoop claims data between 2011−2021 are shown (Fig 2). There are a total of 30 million service dates, 34.6 million diagnosis codes, 17.4 million procedure codes, and 30.7 million national drug codes observed in AoU data for the 251,664 matched patients with EHR data. In comparison, Swoop found 32.4 (+2.4) million service dates, 36.4 million (+1.8) diagnosis codes, 39.6 (+22.1) million procedure codes, and 20.5 (−10.2) million drug codes. The counts of service dates and diagnosis codes are relatively equal between the two sources, but procedure codes are much higher in Swoop data, while drug codes are higher in AoU data. When combining the total number of PEMs (adding Dx, Px, and NDCs), Swoop (96.4 million) has roughly 17% more activity counts compared to AoU (82.7 million) for matched patients.

### Analysis of recorded healthcare activity in EHR vs claims in shared vs non-shared months

In some months both AoU EHR data and Swoop claims report healthcare activity for any given patient while in other months only one of the two sources report activity. We broke down PEMs into months in which both AoU and EHR data report healthcare activity and months in which only one of the two data sources report information (Fig 3). When looking

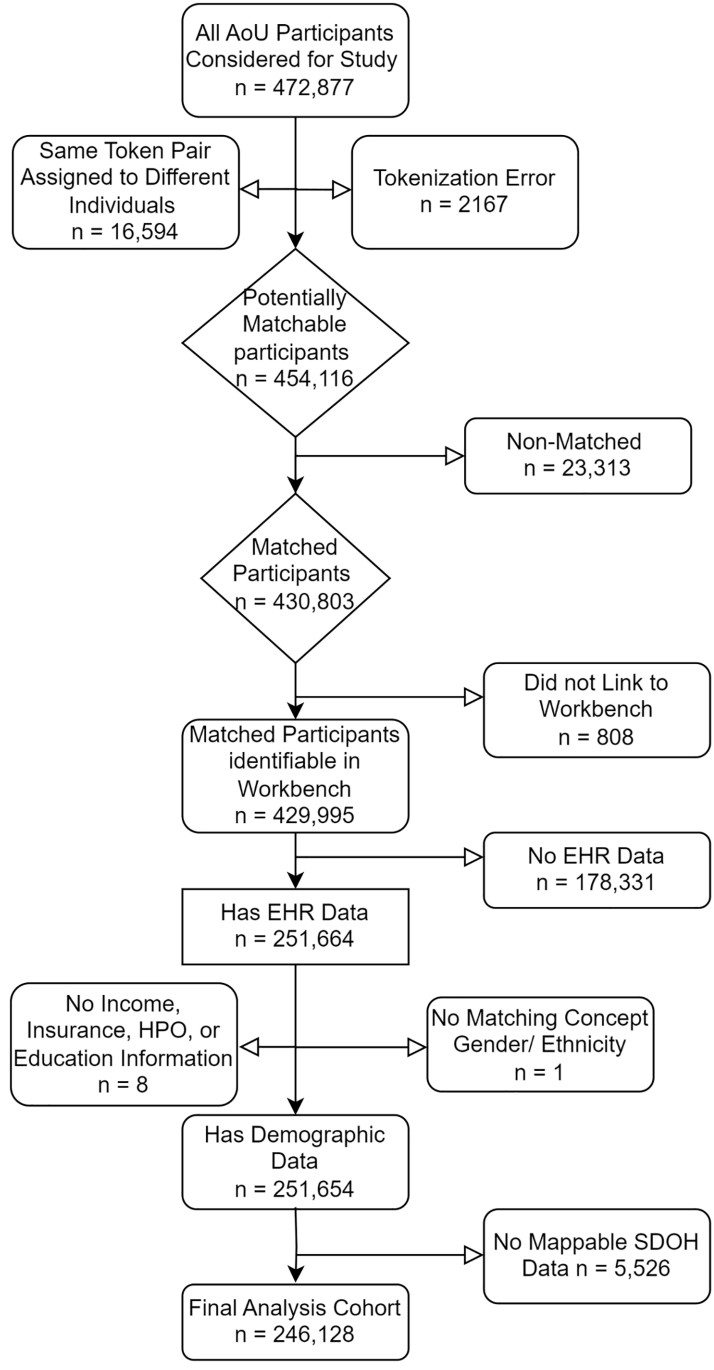

**Fig 1. Breakdown of all participants considered for the study including the number of participants excluded for missing data.** HPO = Healthcare Provider Organization, SDOH = Social Determinants of Health.

at months in which only AoU or only Swoop report activity, it is clear than both data sources provide significant amounts of unique data. Much of the information comes from months in which the two data sources do not overlap. 106.5 million (59.4%) counts of activities are reported in months unique to one of the two data sources, while 72.7 million (40.6%)

**Table 1. Demographic variable breakdown for matched participants in AoU workbench pre and post exclusion of participants without EHR data.**

| N = 375,739 | All Patients with Demographic Data | | N = 246,128 | All Patients with Demographic, SDOH, and EHR data | |
| --- | --- | --- | --- | --- | --- |
| Age | Count (Pre-Filter) | Percentage | | Count (Post-Filter) | Percentage |
| 18-40 | 84879 | 22.590 | | 50265 | 20.422 |
| 41-60 | 119258 | 31.740 | | 76195 | 30.957 |
| 61-80 | 146924 | 39.103 | | 101554 | 41.261 |
| 80+ | 24678 | 6.568 | | 18114 | 7.360 |
| Gender | | | | | |
| Female | 226,859 | 60.377 | | 149653 | 60.803 |
| Male | 138,383 | 36.830 | | 90402 | 36.730 |
| Skips and other responses (Self-Identified) | 10,497 | 2.794 | | 6073 | 2.467 |
| Race | | | | | |
| White | 211826 | 56.376 | | 136692 | 55.537 |
| Black | 71146 | 18.935 | | 48419 | 19.672 |
| Asian | 12644 | 3.365 | | 6779 | 2.754 |
| Multiracial | 7174 | 1.909 | | 4369 | 1.775 |
| Other | 6635 | 1.766 | | 4235 | 1.721 |
| Unknown | 66314 | 17.649 | | 45634 | 18.541 |
| Ethnicity (Self-Identified) | | | | | |
| Not Hispanic/Latino | 300741 | 80.040 | | 195018 | 79.234 |
| Hispanic/Latino | 64831 | 17.254 | | 44669 | 18.149 |
| Unknown | 10167 | 2.706 | | 6441 | 2.617 |

SDOH = Social Determinants of Health.

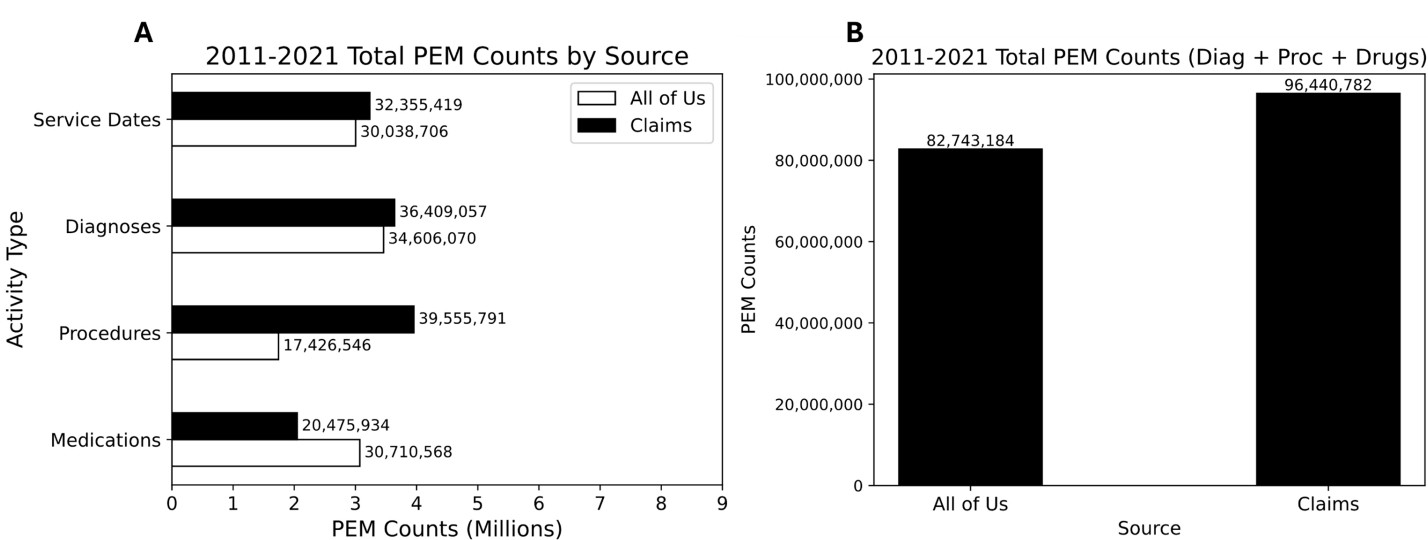

**Fig 2. A) The overall number of PEM between AoU and Swoop data by category. B) The total count of healthcare activities (Dx + Px + NDCs) between AoU and Swoop.**

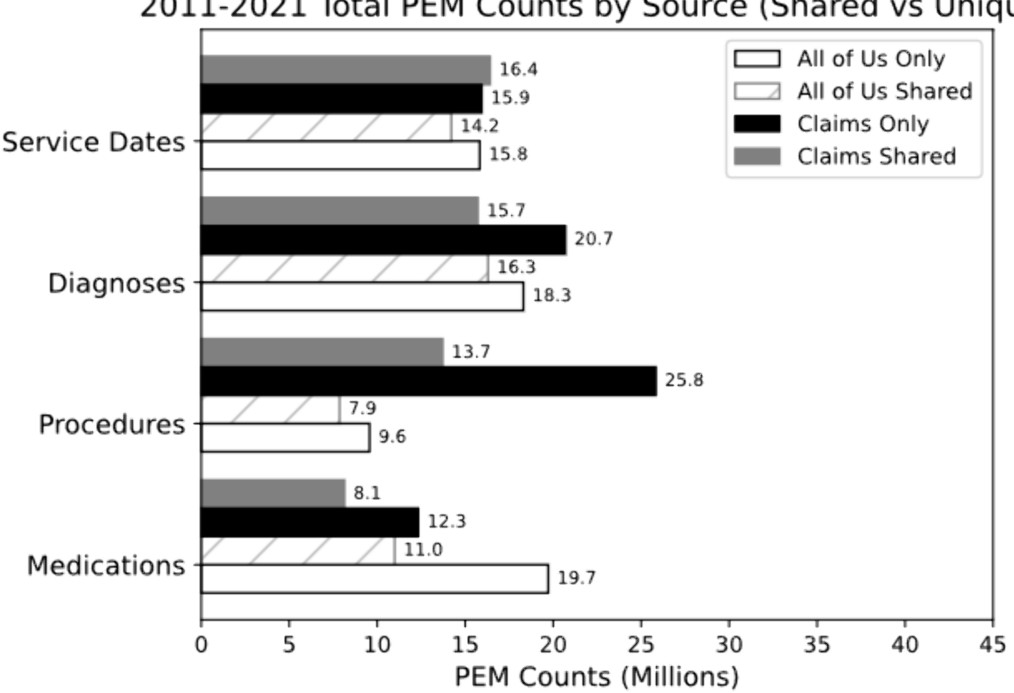

**Fig 3. The amount of All of Us and Swoop PEMs broken down by source in months in which activity is reported by both or only one dataset.** White bar = PEMs in months recorded only by All of Us, White striped bar = PEMs in months in which both All of Us and Claims record activity within All of Us data, Grey Bar = PEMs in months in which both All of Us and Swoop record activity within Swoop data, Black bar = PEMs in months recorded only by Swoop.

counts occur in months during which both AoU and Swoop observe patient healthcare activity. Examining months in which both AoU (All of Us Shared) and Swoop (Claims Shared) report healthcare activity, we see differences in the number of PEMs reported by each data source despite the activity coming in the same month. While diagnosis code counts in shared months is similar, Swoop reports nearly 16% more service dates, 75% more Px codes and 26% fewer NDCs compared to AoU in months in which both data sources observe patient healthcare activity.

### Relative contributions by data source for four care categories over time

Breaking down PEMs into years, trends in the relative contribution of AoU and Swoop PEMs are observed (S1 Fig) The contribution of PEMs observed in AoU EHR data makes up most of the activity during the beginning of the observation period from 2011 through 2013. Contribution of activity between Swoop and AoU data stabilizes around 2014 and stays similar throughout the rest of the study period.

### Analysis of gain in participant health data using claims for participants without existing EHR data

To see how much data would be gained through the harmonization of Swoop claims data into AoU program data for participants without already existing EHR data, the counts of claims events for the 178,331 participants without existing AoU EHR data was aggregated (Fig 4). Overall, a total of 14.8 million service days, 16 million diagnosis codes, 17.8 million procedure codes, and 9.4 million drug codes could be gained from supplementation of Swoop claims data into AoU participant data for participants that do not currently have EHR data in the researcher workbench. At the per participant level, this amounts to an average of 83 service dates, 90 diagnosis codes, 99 procedure codes, and 53 medications per participant.

**Fig 4. The overall number of PEMs found in Swoop claims data for matched AoU participants that do not have EHR data available (n = 178,331).**

## Adjusted analysis of characteristics associated with differential AoU contributions

We examined the relative richness of AoU EHR data compared to claims data for participants depending on their characteristics using a dependent variable consisting of differences in the count of PEMs observed in AoU vs Swoop data per category in a linear regression model (Diagnoses data as outcome shown in Fig 5, service dates, procedures, and medications shown in S2–S4 Figs). When examining what participant characteristics made them more likely to have enriched data in AoU or Swoop data, several trends emerged.

Participant race had little influence on the likelihood of greater AoU or Swoop-sided information, although Asian participants and participants with Unknown race had slightly higher AoU-sided information throughout PEM categories. Hispanic patients had consistently higher Swoop-sided information in all PEM categories, with an extra 20–33 Swoop counts of activity per category relative to non-Hispanic participants. Participant age correlated almost linearly with greater Swoop-sided data, with higher age increasingly associated with more claims-sided healthcare activity. Individuals who enrolled through a Veterans Association had much greater AoU-sided information compared to participants that enrolled through other HPO types, having large effect sizes of greater than 200 service days and procedures, over 179 NDCs and around 92 diagnoses.

For variables relating to social determinants of health, increasingly lower education attainment and lower income both correlated with greater claims data relative to AoU data. Relative to participants with private insurance, participants without coverage have more AoU information compared to Swoop while those with Medicaid have the most consistently high representation in Swoop information. Individuals in public insurance categories (public, Medicare, Medicaid) generally had slightly higher claims-sided data compared to those with private insurance in categories outside of NDCs, where the effect is mostly equal. In comparison to participants living in an urban center, participants who lived in suburbs had slightly more claims-sided information in categories outside of procedures, while patients that lived in small town/rural and especially large rural settings had more AoU information. Deprivation index had relatively little association with the relative strength of healthcare data, with participants in the most deprived category (q4) having the greatest differential association

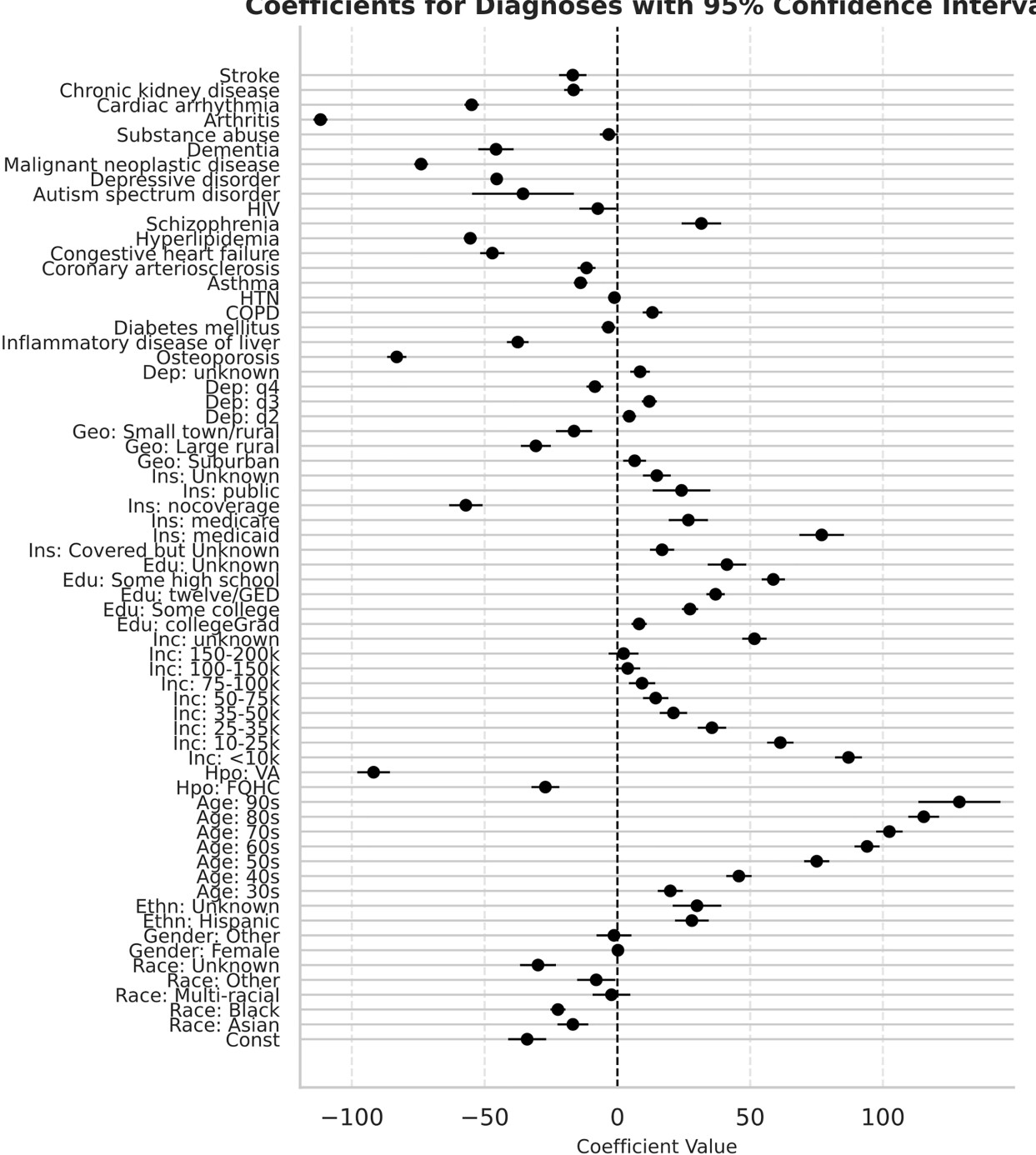

**Coefficients for Diagnoses with 95% Confidence Intervals**

**Fig 5. Adjusted linear regression showing association between patient characteristics and greater presence of diagnoses data within AoU vs Swoop data.** Leftward facing data has stronger AoU contribution while rightward facing data has stronger Swoop contribution. HTN = Hypertension, COPD = Chronic Obstructive Pulmonary Disease Dep: = Deprivation, Geo: = Geography, Ins: = Insurance, Edu: = Education, Inc: = Income, Hpo: = Healthcare Provider Organization, VA = Veterans Association, FQHC = Federally Qualified Health Center, Ethn: = Ethnicity.

compared to participants in q1, with higher (+25) AoU services days, diagnoses (+8), and procedures (+15), and some-what more Swoop NDCs (+12).

The presence of most chronic conditions was associated with higher AoU-sided information in categories other than procedures, where the presence of several comorbid conditions was associated with higher Swoop-sided data. The presence of a few chronic conditions indicated greater Swoop activity. HIV was the only condition with claims-sided medication data, and COPD and especially Schizophrenia had greater Swoop activity in categories outside of medications.

## Discussion

Our study found significant benefit towards data coverage when combining EHR data of AoU participants with their respective claims data. Compared to AoU EHR data, Swoop claims contained roughly 17% more counts of healthcare activity in PEMs overall for matched patients. 59.4% (106.5 million) of total PEMs in the study were found in months that were unique to AoU or Swoop data, suggesting that each data source contains large amounts of information that is not found in the other. The distinctive data of each data asset highlights the need for integrating additional data sources into AoU. Indeed, through the AoU Center for Linkage and Acquisition of Data (CLAD), claims is a data stream being acquired and curated to fill in missingness and enhance research [32]. Additionally, a significant portion of AoU participants have no EHR data recorded, with these participants having the most to gain from linkage to claims (an average of 83 service dates, 90 diagnosis codes, 99 procedure codes, and 53 medications per patient).

We also found differences in the relative strength of the two data sources. Differences in PEM counts in shared months suggest that each dataset is richer in different types of data. Compared to claims, EHR data in AoU has notably increased representation in NDC type information, while claims data is much richer in Procedure codes. This may reflect differences in how data is captured between EHR and claims data. For example, non-prescription medications may not show up in claims data, leading to a higher NDC count in EHRs [33]. Additionally, comparing AoU data to Swoop by year, we see that there is a relative lack of Swoop claims data in 2011 and 2012 compared to later years (S1 Fig).

Our linear regression analysis found several patient factors that are associated with more AoU or sided healthcare activity. Generally, lower income, older, and less educated participants, participants on Medicaid, and participants self-identifying as Hispanic ethnicity had increased claims-sided representation. Participants recruited from the Veterans Association, participants without insurance, and participants living in more rural settings had higher AoU-sided information. The especially striking difference in Veterans Association participants compared to other HPOs may suggest better data coverage in those organizations or decreased tendency to seek care at other institutions. Meanwhile, the greater EHR data coverage in rural participants may be related to increased care options in urban settings leading to higher potential for fragmented care. Large effect sizes towards claims-sided data in older, lower income participants with decreased education attainment, as well as participants self-identifying as Hispanic, may be important considerations for AoU to consider for ensuring data quality in a diverse and equitable way.

Area-level measures of social determinants of health were weakly associated with relative strength of healthcare utilization data in this study, with only individuals in the highest deprivation areas (q4) having somewhat greater AoU-sided information compared to individuals in the least deprived areas. The presence of most comorbid conditions in the EHRs of AoU participants were associated with greater AoU-sided information. In contrast, AoU participants with COPD and especially schizophrenia instead had greater Swoop representation in categories outside of NDCs, while participants with HIV had greater claims information related to medications. The general AoU-sidedness of comorbidity data may be due to existence of EHR data serving as a marker of high information coverage in the EHRs of these patients. The reason for the reversal of this trend with certain comorbidities such as schizophrenia is unknown and may have something to do with increased tendency to have fragmented care in these patients, requiring further investigation.

Improving the quality of EHR data is a major focus of the *All of Us* Research Program. This work offers the first comprehensive look into the extent of data missingness in AoU EHR data and the potential that linkage to ancillary resources

like health insurance claims would provide for program data. Overall, the existence of substantial amounts of information in months covered only in AoU or Swoop, the differential data quantity in shared months in different PEM categories, and the potential gain in healthcare data for participants without EHR provide strong evidence that the AoU EHR data would benefit greatly from linkage to health insurance claims. Deficiencies in relative coverage of AoU EHR data compared to claims we find here based on patient-facing characteristics are important considerations for the AoU research program in ensuring equitable data quality. Further, other factors not considered in this analysis such as relative coverage of EHR data coming from individual HPO enrollment sites may represent useful quality improvement checks for the program in improving data ingestion pipelines. Recently, AoU established the Center for Linkage and Acquisition of Data (CLAD) in order to improve missingness in participant EHR data through linkage to other sources of data [34]. The findings of this work will provide valuable guidance for EHR enrichment efforts like the CLAD to aid AoU in creating a more comprehensive data environment to advance health research.

## Limitations

Our data partner Swoop has claims records for 300 + million unique patient journeys [25]. While our match rate suggests that Swoop does contain data for a large proportion of Americans, it is difficult to know what proportion of total healthcare activity for patients is captured by Swoop's data and what blind spots exist within that data coverage. For one, the analysis in S1 Fig shows that Swoop has gaps in its representation of claims from earlier years, with data contribution between AoU and Swoop evening out around 2013. Per our colleagues at Swoop, the depth of data availability Swoop has for patients is dependent on the extent to which patient's insurance providers make data available in the market for Swoop to use. While comprehensive, the dataset has some limitations worth noting: (1) it may underrepresent certain populations, particularly those who are uninsured or underinsured; (2) data coverage varies by geographic region and insurance type; (3) claims data inherently captures only billable healthcare encounters, potentially missing care provided through non-traditional channels or cash payments; and (4) clinical details beyond what is required for reimbursement are typically not captured. Furthermore, patients receiving clinical services from retail pharmacies with which Swoop or its data vendors have a data sharing agreement with will not appear in Swoop's dataset.

A large proportion of the initially considered 472,877 participants were excluded, mostly due to lack of AoURP EHR data availability, failure to match participants between claims and EHR data, or errors in the tokenization process itself (Fig 1). While we did not identify major differences in participant matching by demographics (Table 1), we acknowledge that hidden biases in AoURP EHR data availability and Swoop claims coverage may impact the study conclusions. An additional limitation comes from our agreement with Swoop which only allowed us to analyze data at the level of the PEMs, restricting the granularity at which we can examine the data. Since we are unable to compare PEMs at the daily level, we are unable to account for date shifts in recorded activity between EHR data capture and claims records (although per Swoop, minor date are infrequent and occur at constant rates; losses across a month boundary are balanced by gains from the preceding month, so monthly PEM totals remain effectively unchanged), which may impact the accuracy of shared vs unique months (Fig 3). This limitation also means we are unable to investigate exactly what codes are driving the differences seen in AoU and Swoop data due to being unable to see healthcare activity data on the Swoop side. In addition to the limitation of PEM resolution in analyzing our data, fundamental differences in the way healthcare events are captured between claims and EHR data may complicate the analysis of our results. Another limitation of this research is that the PDR data used in this activity is not generally available to researchers. Nonetheless, it was used because it is a critical resource to guide programmatic activities in the investigation of data missingness and linkage. Finally, this research points to the gaps inherent to both EHR and claims for research purposes. While EHR may contain lab values and narrative data which is missing in claims, conversely, claims may fill in important areas of missingness such as procedural information. Taken together these two sources are complementary and amplify research.

## Supporting information

**S1 Fig. The relative proportion of patient event months per data source per year.**
(TIF)

**S2 Fig. Adjusted linear regression showing association between patient characteristics and relative strength of service dates data between AoU vs Swoop.** Leftward facing data has stronger AoU contribution while rightward facing data has stronger Swoop contribution. HTN = Hypertension, COPD = Chronic Obstructive Pulmonary Disease Dep: = Deprivation, Geo: = Geography, Ins: = Insurance, Edu: = Education, Inc: = Income, Hpo: = Healthcare Provider Organization, VA = Veterans Association, FQHC = Federally Qualified Health Center, Ethn: = Ethnicity.
(TIF)

**S3 Fig. Adjusted linear regression showing association between patient characteristics and relative strength of procedures data between AoU vs Swoop.** Leftward facing data has stronger AoU contribution while rightward facing data has stronger Swoop contribution. HTN = Hypertension, COPD = Chronic Obstructive Pulmonary Disease Dep: = Deprivation, Geo: = Geography, Ins: = Insurance, Edu: = Education, Inc: = Income, Hpo: = Healthcare Provider Organization, VA = Veterans Association, FQHC = Federally Qualified Health Center, Ethn: = Ethnicity.
(TIF)

**S4 Fig. Adjusted linear regression showing association between patient characteristics and relative strength of medications data between AoU vs Swoop.** Leftward facing data has stronger AoU contribution while rightward facing data has stronger Swoop contribution. HTN = Hypertension, COPD = Chronic Obstructive Pulmonary Disease Dep: = Deprivation, Geo: = Geography, Ins: = Insurance, Edu: = Education, Inc: = Income, Hpo: = Healthcare Provider Organization, VA = Veterans Association, FQHC = Federally Qualified Health Center, Ethn: = Ethnicity.
(TIF)

**S1 Table. Table showing effect sizes of regression coefficients used for linear regression analysis between patient characteristics and relative strength of diagnosis data between AoU vs Swoop.** Negative coefficients represent higher AoU contribution while positive coefficients represent stronger Swoop contribution. HTN = Hypertension, COPD = Chronic Obstructive Pulmonary Disease, VA = Veterans Association, FQHC = Federally Qualified Health Center.
(CSV)

**S2 Table. Table showing effect sizes of regression coefficients used for linear regression analysis between patient characteristics and relative strength of service dates data between AoU vs Swoop.** Negative coefficients represent higher AoU contribution while positive coefficients represent stronger Swoop contribution. HTN = Hypertension, COPD = Chronic Obstructive Pulmonary Disease, VA = Veterans Association, FQHC = Federally Qualified Health Center.
(CSV)

**S3 Table. Table showing effect sizes of regression coefficients used for linear regression analysis between patient characteristics and relative strength of procedures data between AoU vs Swoop.** Negative coefficients represent higher AoU contribution while positive coefficients represent stronger Swoop contribution. HTN = Hypertension, COPD = Chronic Obstructive Pulmonary Disease, VA = Veterans Association, FQHC = Federally Qualified Health Center.
(CSV)

**S4 Table. Table showing effect sizes of regression coefficients used for linear regression analysis between patient characteristics and relative strength of medication data between AoU vs Swoop.** Negative coefficients represent higher AoU contribution while positive coefficients represent stronger Swoop contribution. HTN = Hypertension, COPD = Chronic Obstructive Pulmonary Disease, VA = Veterans Association, FQHC = Federally Qualified Health Center.
(CSV)

## Acknowledgments

We gratefully acknowledge *All of Us* Research Program participants for their contributions, without whom this research would not have been possible. We also thank the National Institutes of Health's *All of Us* Research Program for making available the participant data examined in this study. We would also like to acknowledge Swoop for their support in the development of this manuscript.

## Author contributions

**Conceptualization:** Yuyang Yang, Lew Berman, Melissa Basford, Abel Kho.

**Data curation:** Yuyang Yang, Kelsey Rodriguez, Javier Ezcurra, Romain Bogaerts, Andres Corrada-Emmanuel, Melissa Basford, Abel Kho.

**Formal analysis:** Yuyang Yang, Javier Ezcurra, Romain Bogaerts, Andres Corrada-Emmanuel.

**Funding acquisition:** Lew Berman, Abel Kho.

**Investigation:** Yuyang Yang, Javier Ezcurra, Romain Bogaerts, Andres Corrada-Emmanuel, Abel Kho.

**Methodology:** Yuyang Yang, Javier Ezcurra, Romain Bogaerts, Andres Corrada-Emmanuel, Lew Berman, Abel Kho.

**Project administration:** Kelsey Rodriguez, Lew Berman, Melissa Basford.

**Resources:** Kelsey Rodriguez, Javier Ezcurra, Romain Bogaerts, Andres Corrada-Emmanuel, Melissa Basford.

**Software:** Romain Bogaerts, Andres Corrada-Emmanuel.

**Supervision:** Andres Corrada-Emmanuel, Lew Berman, Melissa Basford, Abel Kho.

**Validation:** Yuyang Yang, Andres Corrada-Emmanuel.

**Visualization:** Yuyang Yang, Javier Ezcurra, Andres Corrada-Emmanuel.

**Writing – original draft:** Yuyang Yang, Kelsey Rodriguez, Javier Ezcurra, Romain Bogaerts, Lew Berman, Melissa Basford, Abel Kho.

**Writing – review & editing:** Yuyang Yang, Kelsey Rodriguez, Javier Ezcurra, Romain Bogaerts, Lew Berman, Melissa Basford, Abel Kho.

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
