## [Decision Letter · Decision Letter 0]

8 Jul 2025

Dear Dr. Yang,

Thank you for submitting your manuscript to PLOS ONE. After careful consideration, we feel that it has merit but does not fully meet PLOS ONE’s publication criteria as it currently stands. Therefore, we invite you to submit a revised version of the manuscript that addresses the points raised during the review process.

We look forward to receiving your revised manuscript.

Kind regards,

Sreeram V. Ramagopalan

Academic Editor

PLOS ONE

Journal Requirements:

“Abel Kho is an advisor of the company Datavant.”

We note that one or more of the authors are employed by a commercial company: Datavant

4. Please note that your Data Availability Statement is currently missing [the repository name and/or the DOI/accession number of each dataset OR a direct link to access each database]. If your manuscript is accepted for publication, you will be asked to provide these details on a very short timeline. We therefore suggest that you provide this information now, though we will not hold up the peer review process if you are unable.

5. Please remove all personal information, ensure that the data shared are in accordance with participant consent, and re-upload a fully anonymized data set.

Reviewers' comments:

Reviewer's Responses to Questions

**Comments to the Author**

1. Is the manuscript technically sound, and do the data support the conclusions?

Reviewer #1: Yes

Reviewer #2: Partly

2. Has the statistical analysis been performed appropriately and rigorously?

Reviewer #1: Yes

Reviewer #2: I Don't Know

3. Have the authors made all data underlying the findings in their manuscript fully available?

Reviewer #1: Yes

Reviewer #2: No

4. Is the manuscript presented in an intelligible fashion and written in standard English?

Reviewer #1: Yes

Reviewer #2: Yes

Reviewer #1: This manuscript presents a timely and relevant investigation into the completeness of EHR data within the All of Us Research Program, through linkage with data provided by Swoop. The scale of the dataset and the analytical approach are commendable and contribute meaningfully to the field of data integration in population health research.

Suggestions for Improvement

The introduction would benefit from a clearer explanation of what Swoop is, how it obtains its data, and the scope and limitations of its claims dataset. This context is essential for readers to assess the representativeness and reliability of the claims data used in the study.

The manuscript should more explicitly explain the structural and functional differences between EHR and claims data. This includes their respective purposes, typical content, and known limitations. Such clarification would help readers understand why certain types of information are more prevalent in one source than the other.

It would be helpful to briefly discuss why EHR and claims data are not routinely integrated in the U.S. healthcare system, touching on technical, legal, and institutional barriers. This would provide important context for the significance of the authors’ linkage efforts.

The manuscript states that 472,877 participants were considered for linkage, but only 246,128 were included in the final analysis. This represents a substantial reduction that should be clearly stated.

The authors should estimate and report the volume of diagnostic, procedural, and medication data that was not utilized due to unmatched or excluded participants. These figures should be critically discussed in terms of potential data loss, bias, and the impact on the study’s conclusions.

While the manuscript includes chronic conditions as covariates in regression models, it does not explore the completeness of data for these subgroups in detail. Given the importance of chronic disease populations in health research, the authors are encouraged to provide more granular analysis or discussion on how data completeness varies by condition. This would be particularly valuable for assessing equity and data quality across clinically vulnerable groups.

The percentage values in Table 1 currently display inconsistent decimal places. For clarity and professionalism, it is recommended that all percentages be presented with a uniform number of decimal places.

Reviewer #2: This study makes a valuable contribution by providing practical insights into maximizing patient-level data availability for the research of secondary use of data. However, several critical points require clarification to strengthen the manuscript's methodological rigor and practical applicability.

Point 1.

The manuscript appears to assume that duplicate data between EHR and claims sources were identified and excluded from analysis, particularly given the use of patient-event months (PEMs) as the unit of comparison. However, the methods lack a clear definition of what constitutes a duplicate event across databases (e.g., same patient, same month, same code), nor do they detail how such duplicates were detected and managed.

How were duplicate events defined between EHR and claim data? What criteria were used to identify duplicate diagnoses, procedures, and medications across EHR and claims data? How were potential legitimate duplicates (e.g., bilateral procedures, multiple prescriptions) distinguished from true redundancies?

Point 2

While the manuscript demonstrates the quantitative gain in data elements through claims linkage, it does not sufficiently discuss the specific types of research questions or clinical scenarios where claims data integration is most impactful. The authors should elaborate on concrete use cases where the breadth and continuity of claims data provide unique advantages over EHR data alone. Articulating these scenarios would help readers understand the practical value of data expansion in the research using secondary use of data.

Point 3

The study equates an increase in the number of captured data elements with improved “completeness.” In data quality frameworks, completeness refers to the absence of missing values within captured records, not the comprehensiveness of data sources. Clearly distinguish between: (a) completeness (non-missing values in existing records), (b) coverage (proportion of all healthcare events captured), and (c) comprehensiveness (breadth of data sources/types)

Point 4

The manuscript provides insufficient detail about how heterogeneous data from EHR and claims sources were harmonized for comparison. Given the inherent differences in data structure, coding systems, temporal granularity, and semantic representations between EHR and claims data, the manuscript should clearly describe the harmonization process used to enable valid comparisons. What mapping strategies, code translations, or aggregation rules were applied to ensure that events from both sources were comparable at the PEM level? Were there limitations in mapping certain codes or event types? How were discrepancies in coding standards addressed? Transparent reporting of these harmonization steps is essential for reproducibility and for interpreting the validity of the comparative analyses.

Point 5

The manuscript clearly states that the study received approval from the relevant Institutional Review Boards and that written informed consent was obtained from participants. Furthermore, the process of privacy-preserving record linkage using Datavant software—whereby personal identifiers are tokenized and encrypted prior to linkage—appears to be well described. The authors also note that researchers did not have access to identifiable information at any stage, and that all analyses were conducted within a secure research environment. However, I respectfully suggest that the manuscript could be strengthened by providing additional detail regarding the specific content of the participant consent forms, particularly regarding the scope of data linkage and secondary data use. Additionally, could you elaborate on the legal framework supporting this data linkage?

**Do you want your identity to be public for this peer review?** For information about this choice, including consent withdrawal, please see our Privacy Policy

Reviewer #1: **Yes:** António da Luz Pereira

Reviewer #2: No

---

## [Author Response · Author response to Decision Letter 1]

13 Aug 2025

- We have reviewed the style templates and have adjusted figures/tables/in-text citations to reflect these guidelines.

- Additional information about consent has been included in the ethics statement and Methods section.

“Abel Kho is an advisor of the company Datavant.”

We note that one or more of the authors are employed by a commercial company: Datavant

- An updated funding statement and competing interest statement have been amended to the cover letter

4. Please note that your Data Availability Statement is currently missing [the repository name and/or the DOI/accession number of each dataset OR a direct link to access each database]. If your manuscript is accepted for publication, you will be asked to provide these details on a very short timeline. We therefore suggest that you provide this information now, though we will not hold up the peer review process if you are unable.

- This link is now provided (https://workbench.researchallofus.org/workspaces/aou-rw-f6117030/duplicateofevaluatevisitspermonth/about)

5. Please remove all personal information, ensure that the data shared are in accordance with participant consent, and re-upload a fully anonymized data set.

- In order to preserve AoURP program data within the AoURP network (and due to size restriction in upload on the PLOS One portal), we have added a link to our workspace in point 4 for data analysis which contains a fully anonymized dataset.

The introduction would benefit from a clearer explanation of what Swoop is, how it obtains its data, and the scope and limitations of its claims dataset. This context is essential for readers to assess the representativeness and reliability of the claims data used in the study.

- We thank the reviewer for raising this point and agree that additional context surrounding Swoop is needed in the manuscript introduction. We have included new information surrounding Swoop (a company which provides precision healthcare omnichannel solutions) and the source of its claims dataset (multiple primary sources including pharmacy benefit managers, clearinghouses, and payer organizations). We have also included additional information about the limitations of its claims dataset [“(1) it may underrepresent certain populations, particularly those who are uninsured or underinsured; (2) data coverage varies by geographic region and insurance type; (3) claims data inherently captures only billable healthcare encounters, potentially missing care provided through non-traditional channels or cash payments; and (4) clinical details beyond what is required for reimbursement are typically not captured”] which has been added to the limitations section of the study.

The manuscript should more explicitly explain the structural and functional differences between EHR and claims data. This includes their respective purposes, typical content, and known limitations. Such clarification would help readers understand why certain types of information are more prevalent in one source than the other.

We have added an additional section to the introduction (“Use of Insurance Claims to Supplement EHR data”) explaining the role of insurance claims in population health research, and the differences between EHR and claims data.

It would be helpful to briefly discuss why EHR and claims data are not routinely integrated in the U.S. healthcare system, touching on technical, legal, and institutional barriers. This would provide important context for the significance of the authors’ linkage efforts.

-Context has been added into barriers responsible for difficulty integrating EHR and claims data within “Use of Insurance Claims to Supplement EHR data”.

The manuscript states that 472,877 participants were considered for linkage, but only 246,128 were included in the final analysis. This represents a substantial reduction that should be clearly stated.

-We have adjusted this language in the material and methods section to reflect the final analysis population rather than the initial considered participants. Explicit mention of the percentage reduction is added to the “Linkage Evaluation” Section. Language surrounding the reason behind why participants did not have EHR data available within AoURP data (because participants did not consent to release their EHR information, a lag of EHR information ingestion from partner healthcare provider organizations into AoURP, or online enrollment of patient without provided EHR information), is added in the “Linkage Evaluation” section of the text.

The authors should estimate and report the volume of diagnostic, procedural, and medication data that was not utilized due to unmatched or excluded participants. These figures should be critically discussed in terms of potential data loss, bias, and the impact on the study’s conclusions.

- We appreciate this point by the reviewer and agree that understanding the reasons behind potential data loss and the impact of data loss on the study conclusions is important. To understand data loss properly, we want to break down the major sources of data loss described in Figure 1.

1. A large majority of participants not included in our study were filtered out due to lack of EHR data in the AoURP platform (n =178,331). This is due to participants not providing a HIPAA authorization to allow sharing of their EHR data with the program, a lag in the receipt of participant EHR from partner Health Provider Organizations or that the participants joined the program digitally with no available EHR data. Similarly, data loss due to lack of SDOH data within AoURP data (n=5,526) is also due to participants not filling out surveys from which this SDOH data can be sourced.

2. A smaller proportion of participants were excluded due to an issue linking the participants EHR and claims data (n = 23,313). The sources of non-matching during this step can arise from two sources 1) the AoURP participant information is not contained within Swoop’s claims data or 2) Differences in First/Last name, sex at birth, and date of birth between the two datasets vary, as these are the characteristics used to generate Token 1 and Token 2 during matching. To assess potential bias in matching, we include Table 1 which shows that the relative patient demographic breakdown pre and post exclusion are relatively similar, suggesting that data loss due to lack of match between EHR and claims data does not seem to localize to a particular group (at least from a demographic standpoint).

3. The final source of data loss occurred during the tokenization process itself (Same Token Pair Assigned to Different Individuals n = 16,594 or Tokenization Error n = 2167). These errors occur due to 1) There is a lack of fields available from which a token is generated in either input dataset (such as no first name, last name, date of birth) 2) If two or more participants share the exact same inputs from which Token1 and Token2 are generated (have the exact same name and date of birth)

We prefer to include a breakdown of the total number of participants “lost” at each step rather than a breakdown of non-utilized data, as most of these participants fall under condition #1, meaning they do not have EHR data available for which to even report. Additional context behind these sources of data exclusion are added to the manuscript under “Linkage Evaluation” and “Limitations”.

While the manuscript includes chronic conditions as covariates in regression models, it does not explore the completeness of data for these subgroups in detail. Given the importance of chronic disease populations in health research, the authors are encouraged to provide more granular analysis or discussion on how data completeness varies by condition. This would be particularly valuable for assessing equity and data quality across clinically vulnerable groups.

-We thank the reviewer for raising this point and agree that more granular analysis is important in assessing differences in data completeness by patient condition, as well as their causes from a health equity perspective. As the reviewer notes, we do take a preliminary look at this using an adjusted multiple regression model examining some of the most common chronic conditions, patient demographic, as well as the indicators for socio-economic deprivation that were available to us. This manuscript is a primary investigation with two major goals 1) to gain insight into differences in data characteristics between claims and EHR data in a large (~250K), diverse, multi-year retrospective cohort, something that is seldom done on a large scale due to difficulties in data integration between the two sources and 2) to provide internal benefit towards AoURP research userbase and leadership in better understanding strengths and weaknesses of program EHR data, and whether program data would benefit from integration with secondary sources such as claims. While we agree with the reviewer that deeper analysis into why some groups of patients may have higher EHR or claims data in comparison to others, it is not within the primary objectives of this manuscript and we believe that deeper analysis aside from what is provided in Figure 5 represents a substantial effort that is better served as a focus of future work by our group. Further, at this stage we can only speculate on the reasons behind relative deficiencies in claims and EHR data comprehensiveness for patients of different groups. It will require internal investigations by the AoURP based on the initial findings in this paper to better understand causes of these differences and whether they constitute disparities from a health equity standpoint.

The percentage values in Table 1 currently display inconsistent decimal places. For clarity and professionalism, it is recommended that all percentages be presented with a uniform number of decimal places.

-We appreciate the reviewer’s attention in identifying this stylistic inconsistency and have adjusted the percentage values accordingly.

Reviewer #2: This study makes a valuable contribution by providing practical insights into maximizing patient-level data availability for the research of secondary use of data. However, several critical points require clarification to strengthen the manuscript's methodological rigor and practical applicability.

Point 1.

The manuscript appears to assume that duplicate data between EHR and claims sources were identified and excluded from analysis, particularly given the use of patient-event months (PEMs) as the unit of comparison. However, the methods lack a clear definition of what constitutes a duplicate event across databases (e.g., same patient, same month, same code), nor do they detail how such duplicates were detected and managed.

How were duplicate events defined between EHR and claim data? What criteria were used to identify duplicate diagnoses, procedures, and medications across EHR and claims data? How were potential legitimate duplicates (e.g., bilateral procedures, multiple prescriptions) distinguished from true redundancies?

-To clarify, our study design intentionally did not exclude data duplicated between AoU and Swoop datasets, as per agreements between AoURP and Swoop, data analysis would be limited to the PEM level and codes (ICD, NDC, CPT) would not directly be compared between the two organizations. As we mention in our limitations section, this reduces the granularity from which we can identify true unique data coming from either source. Instead, uniqueness is implied by the existence of healthcare interactions occurring on separate days across the two data-sources leading to differences in PEM counts in either dataset. Uniqueness is further supported by analysis in Figure 3, showing the high percentage of PEMs coming from months in which one data source reports events while the other does not.

Swoop has also addressed potential temporal discrepancies of the same duplicate data (such as the same ICD code being reported on a Tuesday in Swoop vs Wednesday in EHR data, leading to false differences in PEM counts between the two datasets). Prior analysis by Swoop confirms that occasional minor date shifts in health records from admini

---

## [Decision Letter · Decision Letter 1]

4 Nov 2025

Investigation into EHR data coverage in the All of Us Research Program via linkage to health insurance claims

PONE-D-25-23520R1

Dear Dr. Yang,

We’re pleased to inform you that your manuscript has been judged scientifically suitable for publication and will be formally accepted for publication once it meets all outstanding technical requirements.

Kind regards,

Sreeram V. Ramagopalan

Academic Editor

PLOS ONE

Additional Editor Comments (optional):

Reviewers' comments:

Reviewer's Responses to Questions

**Comments to the Author**

Reviewer #2: All comments have been addressed

2. Is the manuscript technically sound, and do the data support the conclusions?

Reviewer #2: Yes

3. Has the statistical analysis been performed appropriately and rigorously?

Reviewer #2: Yes

4. Have the authors made all data underlying the findings in their manuscript fully available?

Reviewer #2: Yes

5. Is the manuscript presented in an intelligible fashion and written in standard English?

Reviewer #2: Yes

Reviewer #2: The authors have thoughtfully and thoroughly addressed the concerns raised in the initial review, and the revised version represents a clear advancement in quality. I am pleased to recommend acceptance of the manuscript.

**Do you want your identity to be public for this peer review?** For information about this choice, including consent withdrawal, please see our Privacy Policy

Reviewer #2: No

---

## [Editor Report · Acceptance letter]

PONE-D-25-23520R1

PLOS One

Dear Dr. Yang,

I'm pleased to inform you that your manuscript has been deemed suitable for publication in PLOS One. Congratulations! Your manuscript is now being handed over to our production team.

Kind regards,

on behalf of

Dr. Sreeram V. Ramagopalan

Academic Editor

PLOS One